# Biological Effects of Transforming Growth Factor Beta in Human Cholangiocytes

**DOI:** 10.3390/biology11040566

**Published:** 2022-04-08

**Authors:** Elisa Ceccherini, Nicoletta Di Giorgi, Elena Michelucci, Giovanni Signore, Lorena Tedeschi, Federico Vozzi, Silvia Rocchiccioli, Antonella Cecchettini

**Affiliations:** 1Clinical Physiology Institute-CNR, 56124 Pisa, Italy; digiorgi@ifc.cnr.it (N.D.G.); emichelucci@ifc.cnr.it (E.M.); tedeschi@ifc.cnr.it (L.T.); vozzi@ifc.cnr.it (F.V.); silvia.rocchiccioli@ifc.cnr.it (S.R.); antonella.cecchettini@unipi.it (A.C.); 2Biochemistry Unit, Department of Biology, University of Pisa, 56126 Pisa, Italy; giovanni.signore@unipi.it; 3Department of Clinical and Experimental Medicine, University of Pisa, 56126 Pisa, Italy

**Keywords:** cholangiocytes, TGF-β, proteomics

## Abstract

**Simple Summary:**

Transforming growth factor β (TGF-β) is involved in fibrosis, and contributes to the progressive pathology of cholangiopathies. However, little is known regarding the effects and signaling of TGF-β in cholangiocytes. Here, we assessed the effects of TGF-β on proliferation, cell migration and cell cycle after 24 and 48 h. Proteomic approach was used to highlight proteins involved in these biological processes. In cholangiocytes, TGF-β reduced the proliferation rate and induced cell cycle arrest in G0/G1 phase. Proteomic analysis showed a downregulation of proteins involved in Ca^2+^ homeostasis, including CaM kinase II subunit delta, caveolin-1, NipSnap1 and calumin. Accordingly, Gene Ontology indicated that the plasma membrane and endoplasmic reticulum are the cellular compartments most perturbed following TGF-β treatment. In conclusion, our study highlighted the connection between TGF-β and Ca^2+^ homeostasis in cholangiocytes, and for the first time, correlated calumin and NipSnap1 to TGF-β signaling.

**Abstract:**

TGF-β is a cytokine implicated in multiple cellular responses, including cell cycle regulation, fibrogenesis, angiogenesis and immune modulation. In response to pro-inflammatory and chemotactic cytokines and growth factors, cholangiocytes prime biliary damage, characteristic of cholangiopathies and pathologies that affect biliary tree. The effects and signaling related to TGF-β in cholangiocyte remains poorly investigated. In this study, the cellular response of human cholangiocytes to TGF-β was examined. Wound-healing assay, proliferation assay and cell cycle analyses were used to monitor the changes in cholangiocyte behavior following 24 and 48 h of TGF-β stimulation. Moreover, proteomic approach was used to identify proteins modulated by TGF-β treatment. Our study highlighted a reduction in cholangiocyte proliferation and a cell cycle arrest in G0/G1 phase following TGF-β treatment. Moreover, proteomic analysis allowed the identification of four downregulated proteins (CaM kinase II subunit delta, caveolin-1, NipSnap1 and calumin) involved in Ca^2+^ homeostasis. Accordingly, Gene Ontology analysis highlighted that the plasma membrane and endoplasmic reticulum are the cellular compartments most affected by TGF-β. These results suggested that the effects of TGF-β in human cholangiocytes could be related to an imbalance of intracellular calcium homeostasis. In addition, for the first time, we correlated calumin and NipSnap1 to TGF-β signaling.

## 1. Introduction

Transforming growth factor beta (TGF-β) is a cytokine with multiple effects on cellular processes including proliferation, migration, invasion, angiogenesis and immune responses. One of the well-studied functions of TGF-β is its antimitotic effect, acting through the cell cycle arrest in different cultured cells and mouse models [1]. Although TGF-β possesses cytostatic properties in different cell types, it stimulates the proliferation of endothelial cells and mesenchymal cells, and promotes cell migration in both non-tumor cells and cancers [2,3,4]. TGF-β is involved in liver damage including inflammation, fibrosis and cancer [5]. During hepatic fibrosis, TGF-β regulates the behavior of hepatocytes and the activation of human stellate cells (HSCs) that represent the main source of the extracellular matrix [6,7,8,9,10,11,12,13]. However, the effects of TGF-β on cholangiocytes pathophysiology remain poorly investigated and not well defined. Cholangiocytes are a heterogeneous population of epithelial cells that line the biliary tree. Their major physiologic function is the active modification of bile composition occurring through the modulated transport of different ions, solutes and water across epithelial plasma membranes [14]. In response to injury, cholangiocytes become reactive and acquire a secretory phenotype producing pro-inflammatory and chemotactic cytokines and growth factors that act in paracrine and autocrine manner recruiting inflammatory and mesenchymal cells [15,16]. In addition, Aseem and colleagues have recently defined the mechanisms by which TGF-β signaling mediates the HSC-activating signals from cholangiocytes [17]. The aberrant activity of TGF-β1 signaling has been revealed in a murine model of primary biliary cirrhosis (PBC) [18] and reflected the severity of fibrosis in patients with PBC [19,20], and thus has been proposed as a treatment target in multiple clinical studies [21]. Nowadays, literature data are scarce; thus, it is not clear how TGF-β signaling influences cholangiocytes in physiologic and pathologic conditions. Elucidation of TGF-β effects on this cell type is the first step to yield biliary tree pathogenesis and potential therapeutic targets. For this reason, we assessed the effects of TGF-β on human cholangiocytes at the cellular and molecular levels. In particular, proteomic data suggested that TGF-β could influence Ca^2+^ homeostasis and could inhibit cell proliferation. For the first time, calumin and NipSnap1 have been associated with the TGF-β signaling pathway.

## 2. Materials and Methods

### 2.1. Chemicals and Materials

TGF-β (Sigma-Aldrich/Merck, T7039) was freshly dissolved in PBS (Sigma-Aldrich/Merck, 806544) at 10 μg/mL concentration. Formic acid (5.33002) and ammonium bicarbonate (40,867), both eluent additives for liquid chromatography–mass spectrometry (LC-MS), were purchased, respectively, from Merck (Darmstadt, Germany) and Fluka Analytical (Sigma-Aldrich, St. Louis, MO, USA). Methanol (1.06035) and acetonitrile (1.00029), both hypergrade solvents LiChrosolv for LC-MS, were bought from Merck (Darmstadt, Germany), while sodium deoxycholate (D6750) and albumin from bovine serum (A7906) from Sigma-Aldrich (St. Louis, MO, USA). Reagent A (23,228) and reagent B (1,859,078) for Pierce BCA (bicinchoninic acid) Protein Assay were purchased from Thermo Scientific (Rockford, IL, USA). Iodoacetamide (RPN6302V), dithiothreitol (D1 309.0010) and trypsin-modified sequencing grade (11,418,033,001) were obtained, respectively, from GE Healthcare (Chicago, IL, USA), Duchefa Biochemie (Haarlem, The Netherlands) and Roche (Indianapolis, IN, USA). Spin columns (M1003) and their filters (M2110) were purchased from Mo Bi Tec (Goettingen, Germany) while their VersaFlash spherical C18 stationary phase from Supelco Analytical (Bellefonte, PA, USA). Milli-Q deionized water was filtered on a Millipak filter (0.22 μm, MPGL040001) and purified on an LC-Pak cartridge (C18, LCPAK0001) (all Millipore, Bedford, MA, USA).

### 2.2. Cell Culture and Treatment

Primary human cholangiocytes (catalog number: 36755-12) and their specific complete growth media, with and without serum, were purchased from Celprogen (Torrance, CA, USA). Cells were cultured at 37 °C in a humidified incubator with 5% CO_2_ and underwent starvation (1% FBS) during each treatment. TGF-β was used at the concentration of 10 ng/mL in accordance with data literature [22,23,24,25,26,27].

### 2.3. Cell Counting Assay

In total, 6000 cells per well were seeded in 24-well cell culture plates and cultured overnight. The medium was replaced with serum-free or culture medium supplemented with 1% FBS and incubated for 4 h, 24 h and 48 h. For each time, cells were detached and counted with a hemocytometer.

### 2.4. Cytotoxicity Assay

In total, 2000 cells per well were seeded in a 96-well cell culture plates. After incubation at 37 °C in a humidified atmosphere with 5% CO_2_ for 48 h, the culture medium was replaced by TGF-β (10 ng/mL) diluted with the corresponding culture medium supplemented with 1% FBS. Cells were incubated at 37 °C in a humidified atmosphere with 5% CO_2_ for 24 h and 48 h. Finally, 10 μL of the CCK-8 reagent (Sigma-Aldrich, St. Louis, MO, USA) was added into each well, and OD at 450 nm was measured using a microplate reader (FLUOstar Omega, BMG LABTECH, Ortenberg, Germany) after 2 h incubation at 37 °C.

### 2.5. Cell Cycle Analysis by Flow Cytometry

In total, 6000 cells per well were seeded in 24-well cell culture plates and cultured overnight at 37 °C. The cells were treated with TGF-β (10 ng/mL) [22,23,24,25,26,27] diluted with the corresponding culture medium supplemented with 1% FBS and incubated at 37 °C for 24 h and 48 h. Human cholangiocytes were trypsinized, washed twice with PBS and collected by centrifugation at 900× *g* for 10 min. Each sample was fixed in 1.5 mL of cold 70% ethanol, diluted with 6 ml of PBS and centrifuged at 900× *g* for 10 min. The cell pellets were incubated with 0.3 mL of PBS solution containing 50 μg/mL propidium iodide overnight at 4 °C. Cell cycle distributions were analyzed by measuring DNA content using a flow cytometer.

### 2.6. Wound-Healing Assay

Human cholangiocytes were seeded at the density of 11,000 cells per well in a two-well silicone insert with a defined cell-free gap (Ibidi #81176, Gräfelfing, Germany), incubated overnight at 37 °C and 5% CO_2_, allowing cells to adhere and grow. The silicone insert was removed and the image of the area that remained clear of cells was detected following 24 h and 48 h of treatment with TGF-β (10 ng/mL) using the Leica ICC50 HD digital cameras. The analysis of the wound area was performed with ImageJ.

### 2.7. Sample Preparation for Mass Spectrometry-Based Proteomics

A pellet from about 4 × 10^7^ human cholangiocytes was treated with 1% sodium deoxycholate and sonicated for 5 min (five cycles of 20 s with an interval between cycles of 40 s on ice) and then clarified by centrifugation at 16,000× *g* for 10 min at 4 °C. The bicinchoninic acid assay determined protein concentration by using serum albumin as standard. For each condition, 100 µg of proteins was reduced with dithiothreitol (50 mM, for 30 min, at 65 °C) and alkylated using iodoacetamide (100 mM, 30 min, at 37 °C) in dark conditions. Protein digestion was performed using trypsin (*w*/*w* ratio 1:50) at 37 °C for 24 h. Samples were incubated with 10% trifluoroacetic acid for 10 min at 37 °C to quench the trypsin reaction and remove sodium deoxycholate by acid precipitation. Samples were centrifuged at 16,000× *g* for 10 min and subsequently desalted with Mobicol spin columns equipped with 10 µm pore size filters and filled with VersaFlash C18 spherical 70 Å silica particles. The peptide mixture was lyophilized and dissolved in 5% acetonitrile/95% formic acid to achieve a final peptide concentration of 2 µg/µL before liquid chromatography–tandem MS analysis.

### 2.8. MS Acquisitions: IDA and SWATH-MS

Sample analyses were performed using a micro-HPLC Eksigent Ekspert microLC 200 combined with a Triple TOF 5600 mass spectrometer equipped with a Turbo Ion Spray probe as ion source (all ABSCIEX, Concord, ON, Canada). Five µL of each sample, set in an autosampler at 8 °C, was injected onto a C18 Jupiter column (150 mm × 0.3 mm i.d., 4 µm particle size, 90 Å) thermostated at 30 °C and equipped with a micro trap C18 (10 mm × 0.3 mm) (both Phenomenex, Torrance, CA, USA). The flow rate was set at 5 µL/min and the mobile phases A and B were H_2_O and CHCN_3_, respectively, with 0.1% HCOOH. The elution program was: 0 min, 5% B; 1 min, 5% B; 51 min, 22% B; 51.5 min, 90% B; 53.5 min, 90% B; 54 min, 5% B; and 60 min, 5% B. Chromatographic performances and TOF accuracy were evaluated using an intra-run injection (5 µL) of beta-galactosidase 100 fmol/µL. The mass spectrometer was set in positive ion mode and the operation conditions of the ion source were the following: ion spray voltage floating 5.5 kV, probe temperature 150 °C, curtain gas 25 psi, ion source gas 1 and gas 2, respectively, 30 and 20 psi and declustering potential 100 V. N2, as an inert gas, was used for MS/MS (tandem mass) experiments. For ion library generation (protein identification), samples were analyzed (double injection) with an information-dependent acquisition (IDA) tandem mass spectrometry method based on an MS1 survey scan from which the 20 most abundant precursor ions were selected for subsequent fragmentation (CID = collision-induced dissociation). MS1 survey scans were acquired with a maximum resolving power of 30,000 in a range of 250–1250 *m*/*z*, while MS/MS scans at 25,000 of resolving power in a range of 100–1500 *m*/*z* in a high sensitivity mode. CID experiments were carried out using rolling collision energy automatically calculated according to the *m*/*z* and the charge state of the candidate ion, with a collision energy spread (CES) of 5 V. Precursor ions with a charge state of 1+ were excluded from data-dependent selection. Acquired data were subjected to protein identification using Protein Pilot (ABSCIEX, Concord, ON, Canada) as a probabilistic search tool, using the Homo Sapiens taxon (20,381 entries) on UniProtKB/Swiss-Prot database (release March 2021). For the database search: trypsin was selected as the digestion enzyme with a maximum of 2 mixed cleavages; 50 and 25 ppm error tolerances were used for precursor and fragment ions, respectively; and carbamidomethylation of cysteine residues was selected as fixed modification. In the ion library, 1269 proteins were identified with a 1% critical false discovery rate (FDR). The SWATH (Sequential Window Acquisition of all Theoretical mass spectra) data-independent acquisition (DIA) method was used for protein quantitation. The 6 samples (triple injection) were cyclically acquired with the same MS1 survey scan used for the IDA experiment followed by 50 MS/MS experiments with an *m*/*z* sequential variable range (maximum resolving power of 25,000, high sensitivity mode, rolling collision energy, CES 5 V).

### 2.9. Data Analysis

SWATH files, jointly with the ion library, were processed using PeakView (version 2.1) and MarkerView (version 1.2.1) software (both ABSCIEX, Concord, ON, Canada) to extract the peak areas of all the quantifiable peptides and proteins (1149 proteins quantified). Protein abundances were normalized based on the media total abundance per sample. Each protein fold change (FC) value was calculated as the ratio between the mean expression in TGF-β-treated cells and the mean expression in control cells. Proteins were considered differentially expressed when fold changes were higher than 3 (FC ≤ 1/3 or FC ≥ 3). Functional analysis was conducted through DAVID Bioinformatics Resources 6.8 [28,29]. Enrichment analysis was performed using Fisher’s exact test followed by the Benjamini–Hochberg correction. Three separated analyses were conducted with respect to biological processes (BPs), molecular functions (MFs) and cellular components (CCs). For each ontology, functional terms were grouped according to shared proteins and biological meaning. All experimental cell values are expressed as the mean ± standard deviation. The values between different conditions were analyzed by two-way ANOVA followed by Dunnett’s multiple comparisons test using GraphPad Prism software. Statistical analysis for cell cycle samples was performed with the Kruskal–Wallis test using GraphPad Prism software. *p* < 0.05 was considered statistically significant.

## 3. Results

### 3.1. Biological Effects of TGF-β Treatment

In order to investigate TGF-β stimulation on human cholangiocytes, cells had to be cultured in FBS-depleted media to avoid the potential influences of bovine growth factors. To monitor the impact of serum starvation on cell growth, human cholangiocytes were cultured in serum-free and 1% FBS media. Cells were harvested and counted after 4 h, 24 h and 48 h of incubation. As shown in Figure 1A, human cholangiocytes proliferate with a minimum quantity of FBS, even if at a slower pace than cholangiocytes in the complete medium. Conversely, the cell growth slowed down after 24 h and was badly impaired after 48 h in the serum-free medium. Considering these data, treatments with TGF-β were carried out in medium with 1% of FBS. We analyzed the cytostatic effect of TGF-β on cholangiocyte cells using the CCK-8 assays concluding that TGF-β exerts aspecific cytotoxicity on cholangiocyte cells at the dose of 10 ng/mL. As shown in Figure 1B, no significant cytotoxic effect emerged during the first 24 h. Conversely, modest cytotoxicity appears following 48 h of incubation. Next, we better characterized the effect of TGF-β on the cholangiocyte cell cycle using FACS. The distribution of cells along the cell cycle phases indicated an increase in the cell number in the G0/G1 phase following TGF-β treatment compared with the control cells (Figure 1C). The most marked increase appeared after 48 h of treatment. In keeping with this observation, we also registered a reduction in the cell number in the S phase and mitosis. Our data highlighted that the cytotoxic effect of TGF-β is related to cell cycle arrest in a quiescent state.

Considering the ability of TGF-β in controlling migration in some types of cells [30], we performed a wound-healing assay. Figure 2A shows representative images for cells treated with TGF-β for 24 h and 48 h. For each time step, the wound area was calculated (Figure 2B). In the first 24 h, TGF-β treatment blocked cholangiocyte migration, as shown by the leading edge and the wound area in respect to the control cells. Following 48 h of treatments, TGF-β restored cholangiocyte motility even if the wound area was greater than that of the control cells.

### 3.2. Hypothesis Free Proteomics of TGF-β-Treated Cholangiocytes

Human cholangiocytes were digested with trypsin and peptides were prepared for LC-MS analysis. The data output was searched against a human database and 1149 proteins were identified by at least two unique peptides and quantified. In TGF-β-treated cholangiocytes, a total of 25 proteins were differentially expressed, in respect to the control cells, with a greater than ±3-fold mean (Table 1). Considering the time-dependent TGF-β treatment, data analysis showed 10 upregulated and 15 downregulated proteins following 24 h, and 7 upregulated and 18 downregulated proteins following 48 h treatment. Among them, 4 proteins (parathymosin, proline-rich and coiled-coil-containing protein 1, U4/U6 small nuclear ribonucleoprotein Prp31, palladin) were upregulated for the entire duration of TGF-β treatment, whereas 12 proteins were downregulated.

### 3.3. Functional Annotation of Proteins Enriched in TGF-β-Treated Cholangiocytes

Differentially expressed proteins in TGF-β-treated cells were analyzed with DAVID for functional annotation analysis. GO enrichment analysis was performed with respect to cellular components (CCs), biological processes (BPs) and molecular functions (MFs). As shown in Figure 3A, results highlighted an enrichment of proteins in eight CCs (nucleus, cytosol, plasma membrane, mitochondrion, Golgi apparatus, endoplasmic reticulum, cytoskeleton and extracellular space). Most enriched BPs’ GO terms are associated with gene expression, oxidoreductive processes, cell cycle and calcium homeostasis. Most significantly enriched MFs are protein binding, poly(A)RNA binding, nitric-oxide synthase binding, protein transport activity and catalytic activity. Among them, protein binding is the most representative MFs. To verify if these proteins had already been correlated to cholangiocyte pathophysiology and/or to TGF-β signaling, a Medline search was performed, matching each term with “TGF-β”, “cholangiocytes”, “cholangiopathies”, “bile ducts” and “liver diseases”. Based on our results, we focused our attention on four proteins (CaM kinase II subunit delta, caveolin-1, NipSnap1 and calumin) that were interesting for the regulation of cholangiocyte pathophysiology. For each of them, expression levels at 24 h and 48 h are graphically reported in Figure 3B.

## 4. Discussion

TGF-β signaling pathways are complex cascades that exert multiple effects in a broad range of cell types. Numerous studies have reported the roles of TGF-β in liver pathogenesis, including carcinogenesis and fibrogenesis, acting as pro-fibrogenic factors. The aberrant activation of TGF-β1 signaling has been revealed in a murine model of PBC [18] and reflected the severity of fibrosis in patients with PBC [19,20]. However, the role of TGF-β on biliary ducts has not been adequately investigated. In this study, we elucidated the effects of TGF-β signaling in human cholangiocytes, epithelial cells that dial the biliary tree. We demonstrated that TGF-β treatment results in the reduction in the proliferation rate and cell growth arrest. These results are in line with different studies that elucidated the potent role of TGF-β as an antimitogenic factor [31,32]. Proliferation analysis revealed a non-significant impact at 24 h and a mild reduction in the cholangiocyte proliferation rate at 48 h (Figure 1B). Cell cycle analysis using flow cytometry demonstrated that TGF-β-treated cholangiocytes undergo cell cycle arrest at G0/G1 phase, and the effect is more evident after 48 h of treatment (Figure 1C). Furthermore, cells were blocked in their migration process in the first 24 h of treatment, and re-acquired their migratory phenotype at the second time point (Figure 2A,B). These results are in line with some studies that have elucidated the role of TGF-β as an enhancer of cell motility in non-cancerous cells [30,33].

We used a proteomic approach to understand the possible molecular targets by which TGF-β induces those biological behaviors. Proteomic analysis of TGF-β-treated cholangiocytes revealed a sustained downregulation of proteins involved in Ca^2+^ homeostasis, including CaM kinase II subunit delta, caveolin-1, NipSnap1 and calumin (Figure 3B). Considering the subcellular locations of the downregulated proteins and the results of the GO analysis (Figure 3A), we concluded that TGF-β has a more significant impact on endoplasmic reticulum functionality and cell membrane channels. In physiological conditions, mammalian cells exert a fine regulation between the release of Ca^2+^ from internal stores located in the endoplasmic reticulum and influx from the extracellular space via channels located in the plasma membrane. It is already established that TGF-β stimulates the increase in intracellular cytosolic Ca^2+^ concentration in fibroblasts acting on ryanodine-sensitive channels, which in turn modulates gene expression [34,35,36]. In addition, multiple studies correlated the depletion of intracellular Ca^2+^ [37] and TGF-β signaling in mediating cell accumulation in a G0 state [38,39,40]. Following experimental evidence, our study suggests that TGF-β acts through the impairment of the plasma membrane and endoplasmic reticulum functions, leading to an imbalance of cytosolic Ca^2+^ concentration. The imbalance of Ca^2+^ homeostasis perturbs the expression of proteins linked to the cell cycle, which induces an arrest in the G0/G1 phase. Indeed, our GO analysis highlighted the regulation of gene expression as the most enriched biological process. Among these proteins, about 60% are downregulated and implicated in DNA replication and mRNA splicing. These data are consistent with the results of cell cycle analysis that showed a reduction in cholangiocyte percentage in the S and G2/M phase.

According to our knowledge, this is the first study relating calumin, (responsible for the regulation of Ca^2+^ into the endoplasmic reticulum [41]), and NipSnap1, (an inhibitor of TRPV6 channels [42]), to TGF-β signaling. Calumin is a resident Ca^2+^ binding protein in the endoplasmic reticulum that acts as a sensor of Ca^2+^ concentration within the lumen. Interestingly, embryonic fibroblasts from calumin-null mice showed a marked reduction in Ca^2+^ storage within the endoplasmic reticulum [43]. NipSnap1 is a potent inhibitor of TRPV6 activity acting through the block of TRPV6-mediated Na^+^ and Ca^2+^ flux. Moreover, Nipsnap1 is expressed in (re-)absorptive Ca^2+^ tissues, supporting its involvement in the regulation of the Ca^2+^ balance [42]. In TGF-β-treated cholangiocytes, we observed a downregulation of these proteins, suggesting that TGF-β could act through the impairment of TRPV6 function and the reduction in Ca^2+^ storage within the endoplasmic reticulum, resulting in a deregulation of Ca^2+^ homeostasis. Moreover, considering the role of Ca^2+^ signaling in the modification of primary canalicular bile by cholangiocytes through HCO_3_^−^ secretion [44], TGF-β could play an active role in cholangiopathies and other cholestatic pathologies.

Globally, our findings showed the antimitogenic role of TGF-β in cholangiocyte cells potentially through the perturbation of Ca^2+^ homeostasis acting on the plasma membrane and endoplasmic reticulum, which in turn induces an arrest in the G0/G1 phase. In conclusion, we elucidated for the first time the role of TGF-β in modulating pathways in cholangiocytes. We correlated information at the molecular level, derived from whole-cell proteomic analysis, to anti-mitotic and anti-migratory properties of TGF-β in cholangiocytes. Given the crucial importance of TGF-β-associated pathways in cholangiopathies [20,45,46], these findings could represent a rational base for developing novel therapeutic options.

## 5. Conclusions

In conclusion, our data show that TGF-β inhibits cholangiocyte proliferation and migration, and induces a cell accumulation in G0/G1 phase. Most importantly, our proteomic analysis shows that TGF-β could mediate these biological effects through the perturbation of several proteins involved in Ca^2+^ homeostasis, located at the plasma membrane and endoplasmic reticulum level. Considering the role of Ca^2+^ signaling in the bile production process by cholangiocytes, our results provide useful information to elucidate the role of TGF-β in cholangiopathies and other cholestatic pathologies. Finally, yet importantly, our study is the first relating calumin and NipSnap1 to TGF-β signaling.

## Figures and Tables

**Figure 1 biology-11-00566-f001:**
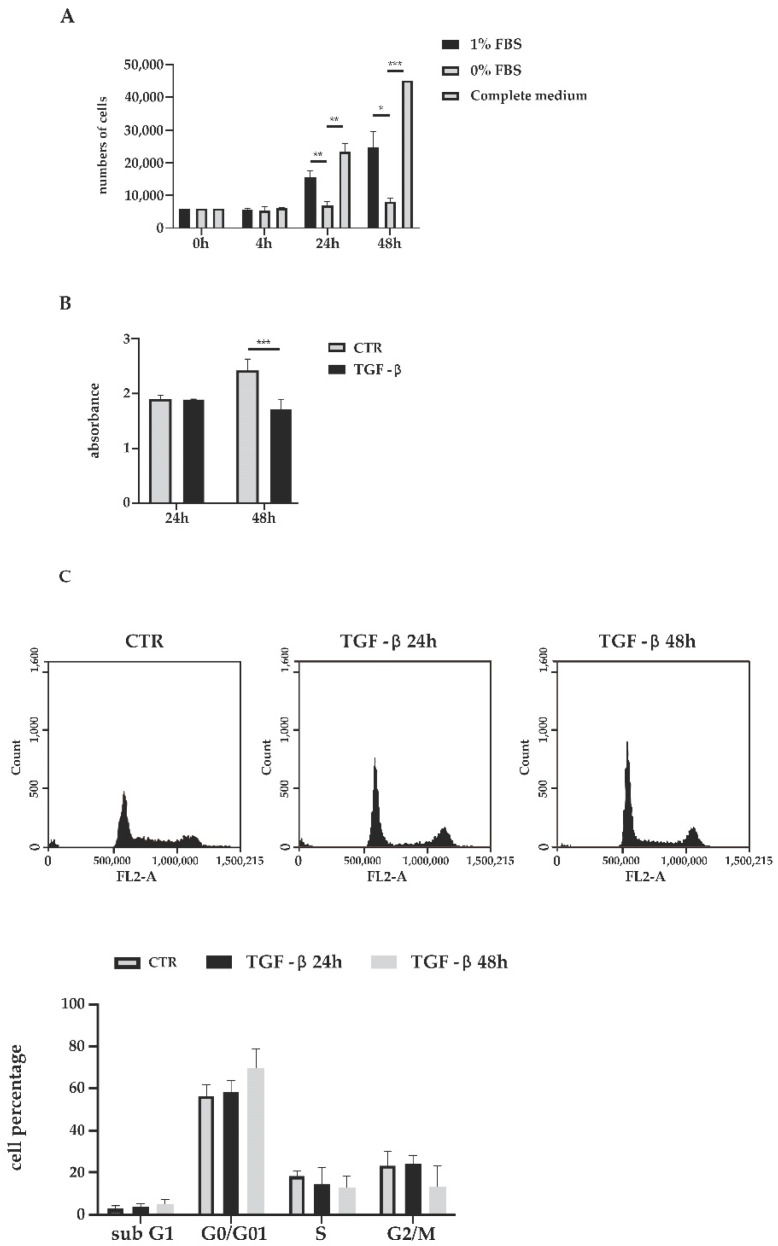
(**A**) Viability of human cholangiocytes exposed to different concentrations of FBS. (**B**) Cytotoxic effect of TGF-β at the dose of 10 ng/mL for 24 h and 48 h. (**C**) Cell cycle analysis by flow cytometry of cholangiocytes cultured in medium with 1% FBS or treated with TGF-β at the dose of 10 ng/mL for 24 h and 48 h. The relative percentage of cells in different cell phases is reported for each time step. Statistical analysis performed with the Kruskal–Wallis test did not show significant differences (*p* < 0.05). * *p* ≤ 0.05, ** *p* ≤ 0.01, and *** *p* ≤ 0.001.

**Figure 2 biology-11-00566-f002:**
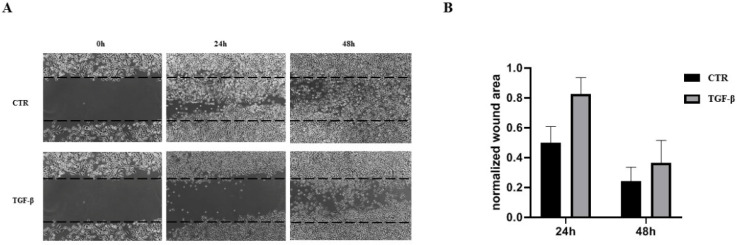
(**A**) Migration assay of cholangiocyte cells treated with TGF-β at the dose of 10 ng/mL compared to the control cells cultured in medium with 1% FBS. (**B**) Bar graphs represent the normalized wound area calculated in two regions of each well at random (*n* = 3). Statistical analysis showed no significant differences between different conditions.

**Figure 3 biology-11-00566-f003:**
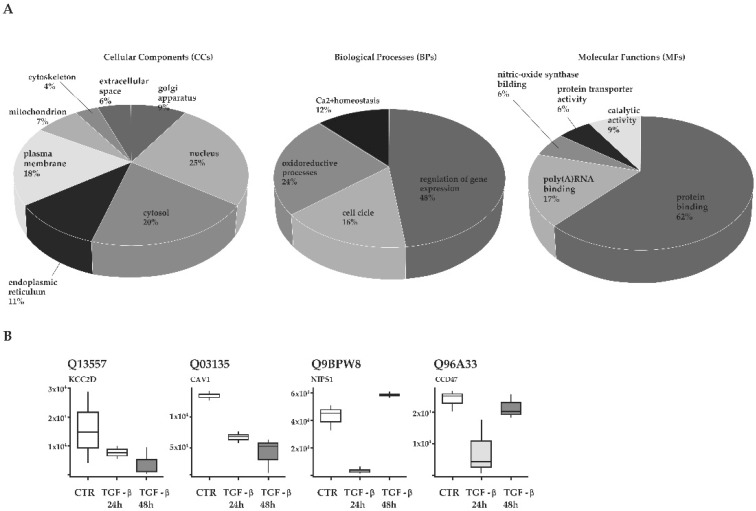
(**A**) GO enrichment analysis of cholangiocytes was retrieved using DAVID. The most enriched GO terms in biological processes (BPs), molecular functions (MFs) and cellular components (CCs) are presented. (**B**) Box plot reporting protein abundances for selected proteins of biological interest in cholangiocyte pathophysiology.

**Table 1 biology-11-00566-t001:** Identified differentially expressed proteins in TGF-β-treated cholangiocytes. The table below reports mean protein abundances and degrees of variation (evaluated as fold change) with respect to the control for TGF-β-treated cholangiocytes at 24 h and 48 h.

Protein Name	Protein ID	CTRL	TGF-β 24 h	TGF-β 48 h	GF-β 24 h vs. CTRL	TGF-β 48 h vs. CTRL
Importin subunit alpha-1	P52292	77,882.7	310,000.0	70,184.0	3.980	0.901
Pre-mRNA-splicing factor SRP55	Q13247	598,000.0	580,333.3	148,666.7	0.970	0.249
Cornifin-A	P35321	4,276,666.7	744,333.3	900,666.7	0.174	0.211
CaMK-II subunit delta	Q13557	15,669.6	7554.6	3596.0	0.482	0.229
Heme oxygenase 1	P09601	64,398.3	282,000.0	74,248.0	4.379	1.153
Parathymosin	P20962	4206.5	13,311.4	27,348.3	3.165	6.501
Caveolin-1	Q03135	133,000.0	65,955.0	41,126.8	0.496	0.309
Prolyl 4-hydroxylase subunit alpha-1	P13674	3170.9	18,216.3	7815.8	5.745	2.465
Thimet oligopeptidase	P52888	3589.1	4644.1	17,872.7	1.294	4.980
Protein NipSnap homolog 1	Q9BPW8	42,351.3	3074.1	58,310.7	0.073	1.377
AP-1 complex subunit beta-1	Q10567	31,677.3	4524.2	28,842.3	0.143	0.911
Albumin	P02768	18,434.3	101,405.3	27,261.3	5.501	1.479
Proline-rich and coiled-coil-containing protein 1	Q96M27	3901.5	13,124.0	26,563.7	3.364	6.809
Glutaredoxin-related protein 5	Q86SX6	44,413.3	13,184.3	24,104.3	0.297	0.543
Eukaryotic initiation factor 4A-II	Q14240	38,164.7	37,378.0	6906.5	0.979	0.181
Replication protein A 14 kDa subunit	P35244	153,937.5	26,327.8	103,743.7	0.171	0.674
Calumin	Q96A33	23,889.7	7707.2	21,445.7	0.323	0.898
Squalene monooxygenase	Q14534	6788.0	29,325.3	2011.4	4.320	0.296
RNA helicase aquarius	O60306	14,418.7	41,652.0	44,619.3	2.889	3.095
ELKS/Rab6-interacting/CAST family member 1	Q8IUD2	6478.8	1507.8	10,435.0	0.233	1.611
U4/U6 small nuclear ribonucleoprotein Prp31	Q8WWY3	2117.3	20,569.7	16,382.3	9.715	7.738
Histidine triad nucleotide-binding protein 1	P49773	9938.9	3275.4	15,506.0	0.330	1.560
Influenza virus NS1A-binding protein	Q9Y6Y0	2593.6	22,095.7	6731.7	8.519	2.596
Palladin	Q8WX93	8038.7	36,757.3	35,210.3	4.573	4.380
Cysteine and glycine-rich protein 2	Q16527	8382.2	19,592.7	26,170.7	2.337	3.122

## Data Availability

The data presented in this study are available on request from the corresponding author.

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
