# Peer review of "Biological Effects of Transforming Growth Factor Beta in Human Cholangiocytes"

_biology, 2022, doi:10.3390/biology11040566_

Round 1
Reviewer 1 Report
The current manuscript entitled "Biological effects of Transforming Growth Factor Beta in human cholangiocytes" authored by Elisa Ceccherini et al; investigated the role of TGF-beta at Calcium homeostasis in primary choalngeocytes. Authors have utilized cell counting and flow methods to access the effects of TGF beta both in complete and serum free media. Identification of differentially expressed proteins was done by quantitative LCMS. I believe that the data is very preliminary and need more experiments to proof their claims.
Comments to authors:
- The quality of all the figures are very poor, it is very hard to read numbers presented in figure 1 C and 3.
- Error bar and statistics are missing in figure 1C, similarly stats are missing in figure 2b.
- Line 199-200, authors claimed that the cytotoxic effect of TGF beta is not mediated by apoptosis, did they perform any experiment to prove the claim?
- Authors have performed quantitative MS to identify differentially expressed proteins post TGF beta induction, what was the internal control and how many times the experiment was repeated. I would also perform validation of identified proteins by western.
- In the discussion line 284-85, authors are hypothesizing that the calcium signaling and concentration is perturbed after the treatment, I don't see any experimental proof of the claim, these kind of claim must be avoided without the proof.
Author Response
- The quality of all the figures are very poor, it is very hard to read numbers presented in figure 1 C and 3.
According to your suggestions, the quality of all the figures was improved, increasing the resolution of the images to 800 dpi.
- Error bar and statistics are missing in figure 1C, similarly stats are missing in figure 2b.
Regarding figure 1C and 2b, the values between different conditions were analyzed by Kruskal–Wallis test and two-way ANOVA followed by Dunnett's multiple comparisons test, respectively, using GraphPad Prism software. P <0.05 was considered statistically significant. Statistical analysis showed no significant differences. We have elucidated these aspects in the captions, as you suggested.
- Line 199-200, authors claimed that the cytotoxic effect of TGF beta is not mediated by apoptosis, did they perform any experiment to prove the claim?
The cell cycle analysis does not show the presence of the sub G1 peak, which is indicative of cell apoptosis. Based on this we say that cytotoxic effect of TGF beta is not mediated by apoptosis.
- Authors have performed quantitative MS to identify differentially expressed proteins post TGF beta induction, what was the internal control and how many times the experiment was repeated. I would also perform validation of identified proteins by western.
Quantitative MS was performed analysing samples with the same protein content. Samples were prepared to achieve a final peptide concentration of 2 µg/µl before liquid chromatography–tandem MS (LC-MS/MS) analysis and 5 µl were injected, thus analysing a total amount of 10 µg of proteins for each sample. In addition, at post-processing data analysis, protein abundances were normalized based on the media total abundance per sample, making the different runs comparable with each other. Chromatographic performances and TOF accuracy were evaluated using an intra-run injection (5 µl) of beta-galactosidase 100 fmol/µl as internal control. LC-MS/MS analyses were repeated in triplicate for each sample.
We agree with the observation of the reviewer about validation by western, but we would underline our choice of LC-MS/MS approach. Sequential Window Acquisition of All Theoretical Mass Spectra (SWATH-MS) is a specific variant of data-independent acquisition (DIA) methods and has developed as a technology that combines deep proteome coverage capabilities with quantitative consistency and accuracy (please refer to following publications for details: L. C. Gillet et al., “Targeted data extraction of the MS/MS spectra generated by data-independent acquisition: A new concept for consistent and accurate proteome analysis,” Molecular and Cellular Proteomics, vol. 11, no. 6, 2012, doi: 10.1074/mcp.O111.016717; C. Ludwig, L. Gillet, G. Rosenberger, S. Amon, B. C. Collins, and R. Aebersold, “ Data‐independent acquisition‐based SWATH ‐ MS for quantitative proteomics: a tutorial ,” Molecular Systems Biology, vol. 14, no. 8, Aug. 2018, doi: 10.15252/msb.20178126).
- In the discussion line 284-85, authors are hypothesizing that the calcium signaling and concentration is perturbed after the treatment, I don't see any experimental proof of the claim, these kind of claim must be avoided without the proof.
We agree with reviewer about the need of an experimental proof. We would like only to say that our hypotheses are based on quantitative proteomic results. These proteomic data can paved the way for studies regarding the evaluation of energy metabolism and calcium homeostasis in TGF‐β-treated cholangiocytes to confirm the hypothesis.

Reviewer 2 Report
The article 'Biological effects of Transforming Growth Factor beta in human cholangiocytes' by Ceccherini et al. proposes investigating the role of TGF‐β stimulation on human cholangiocytes. The area of research is attractive and warrants investigation; however, the manuscript is premature for publication in 'Biology' at this stage. The manuscript does not lay out a reasonable rationale behind the study, and the implications of the results are vague. Further, the authors do not provide relevant functional analyses on their proteomics data. The overall study design is not well thought after, and the data is insufficient. Hence, I do not endorse the publication of the manuscript in 'Biology'. Some of the major concerns limiting the publication are the following:
The rationale behind the proposed study is unclear. The authors discuss in the introduction that TGF‐β signaling has a role in cholangiocyte switching into a reactive and secretory phenotype, thereby producing pro‐inflammatory, chemotactic cytokines, growth factors that can recruit inflammatory and mesenchymal cells. The authors do not attempt to establish the physiological/pathological level of TGF‐β stimulation or score any of the above phenotypes at the physiological or pathological level of TGF‐β stimulation.
No experiments on the major physiological function of cholangiocytes, such as bile composition modification, transport assays, etc., are performed.
No experimentally supported novel information on the physiological/pathological role of TGF‐β in cholangiocytes is reported that is not already known for other systems.
No functional data on proteomic hits such as NipSnap1 or calumin are provided that establishes their roles in TGF‐β stimulated cholangiocyte.
The authors vaguely relate Ca2+ homeostasis to cell proliferation without providing any relevant data in their system.
The discussion is over speculative without convincing data.
The quality of written English can be improved.
Author Response
Thank you for the constructive comments. Introduction and discussion have been thoroughly modified in order to make the rationale clearer and the discussion of the results less speculative and vague. The paper has been carefully revised to improve the English. We hope that in this new form the manuscript can be accepted for publication on Biology.

Reviewer 3 Report
In the present paper, the Authors described the cellular response of human cholangiocytes to TGF-beta treatment. The Authors demonstrated that TGF-beta affected the wound healing process, and this effect of TGF-beta was time-dependent. TGF-beta upregulated and downregulated some proteins as it was estimated by LC-MS/proteomics analysis.
There are several questions:
- Abstract (lines 13-23): The conclusion sentence should be added to the Abstract. The last two sentences of the Introduction are the conclusion, at least, in part. It is not clear why this conclusion was added to the Introduction instead of the Abstract and the Conclusion Section.
- Methods: Line 76. Please, add more details about the cell culture (catalog number; are these cells obtained from the healthy subject; age and gender of the subject if possible).
- Methods: line 95. Please, provide TGF-beta catalog number and the reference(s) for the TGF-beta concentration, used in this study. Was the dose-dependent effect of TGF-beta assessed in this study? The titration of the compound, i.e., investigation of the effect of different concentrations, is recommended.
- Results (lines 207-213): Effect of TGF-beta on wound healing: please, explain the possible mechanism of the opposite effect of TGF-beta at 24-hrs vs. 48-hrs.
- Results, Section 3.2. Please, provide more details for the proteomics data. Please, add the protein expression at both conditions of the TGF-beta treatment of the cholandiocytes, i.e., 24-hrs and 48-hrs to the Table 1. Please, add the folds of expression vs. non-treated cells for 24-hrs and 48-hrs. The Table 1 as it is presented in this version of the manuscript did not provide the full information as it is described in the text (lines 220-229).
- Results, same: please, provide the possible mechanistic basis for the described changes in upregulation and down-regulation of the certain proteins after TGF-beta exposure.
- Figures: the quality of the figures is low, the size of the panels is small; the reading of the graphical results presentation is difficult. I suggest using color where it is possible and improve the quality of the figures.
- Discussion: I suggest breaking the text in the discussion section in several paragraphs for easy reading.
- Discussion: Please, discuss whether the TGF-beta treatment in the present study indeed switched the phenotype of the cholangiocytes as it was stated in the Introduction based on the previous publications (lines 39-44). Was the effect of TGF-beta SMAD-dependent?
- Conclusion: the formal conclusion section is absent. Moreover, the Conclusion paragraph in the end of the Discussion did not fully describe the findings of the present study.
Author Response
- Abstract (lines 13-23): The conclusion sentence should be added to the Abstract. The last two sentences of the Introduction are the conclusion, at least, in part. It is not clear why this conclusion was added to the Introduction instead of the Abstract and the Conclusion Section.
According to your suggestions, we improved the Abstract and added the Conclusion Section.
- Methods: Line 76. Please, add more details about the cell culture (catalog number; are these cells obtained from the healthy subject; age and gender of the subject if possible).
We added the catalog number for human cholangiocytes to the materials section. The manufacturer does not provide the other information requested.
- Methods: line 95. Please, provide TGF-beta catalog number and the reference(s) for the TGF-beta concentration, used in this study. Was the dose-dependent effect of TGF-beta assessed in this study? The titration of the compound, i.e., investigation of the effect of different concentrations, is recommended.
We added the catalog number of TGFbeta, as you suggested. We used the concentration of 10ng/ml as recommended by data literature in case of investigation of TGFbeta biological properties in in vitro models. At this regard, we added more references in order to strengthen our choice.
- Results (lines 207-213): Effect of TGF-beta on wound healing: please, explain the possible mechanism of the opposite effect of TGF-beta at 24-hrs vs. 48-hrs.
Serum starvation is mostly considered as a standard preparatory method in many cellular experiments, including cell migration. However, recent studies give some evidence that serum starvation is a major event that triggers various cell responses and has therefore a great potential to change and interfere with the experimental results. It is documented that serum deprivation causes significant changes in cell migration and also in the expression of migration-related genes. We can assume that in the first 24 hours, cholangiocytes are engaged in adaptation to starvation, and implement compensatory mechanisms whose macroscopic effects are evident in the second part of the treatment. Please refer to following publications for details: “Serum deprivation initiates adaptation and survival to oxidative stress in prostate cancer cells, doi: 10.1038/s41598-020-68668-x”; “Differential migration-related gene expression and altered cytokine secretion in response to serum starvation in cultured MDA-MB-231 cells, doi: 10.1515/abm-2019-0051; Cellular response to oxidative stress: signaling for suicide and survival, doi: 10.1002/jcp.10119).
- Results, Section 3.2. Please, provide more details for the proteomics data. Please, add the protein expression at both conditions of the TGF-beta treatment of the cholandiocytes, i.e., 24-hrs and 48-hrs to the Table 1. Please, add the folds of expression vs. non-treated cells for 24-hrs and 48-hrs. The Table 1 as it is presented in this version of the manuscript did not provide the full information as it is described in the text (lines 220-229).
According to the reviewer suggestions Table 1 in Results, Section 3.2 was revised and all requested data were added.
- Results, same: please, provide the possible mechanistic basis for the described changes in upregulation and down-regulation of the certain proteins after TGF-beta exposure.
Several studies highlighted that TGF-β controls vital cellular functions through its ability to regulate gene expression in normal epithelial cells and in many tumor cells. Considering the shortage of information on the role of TGFb in cholangiocytes, we cannot hypothesize mechanisms of protein regulation using pathways highlighted in other cell types.
- Figures: the quality of the figures is low, the size of the panels is small; the reading of the graphical results presentation is difficult. I suggest using color where it is possible and improve the quality of the figures. According to your suggestions, the quality of all the figures was improved, increasing the resolution of the images to 800 dpi.
- Discussion: I suggest breaking the text in the discussion section in several paragraphs for easy reading.
In order to improve the reading of the manuscript we broken the text in paragraphs, according to your suggestion.
- Discussion: Please, discuss whether the TGF-beta treatment in the present study indeed switched the phenotype of the cholangiocytes as it was stated in the Introduction based on the previous publications (lines 39-44). Was the effect of TGF-beta SMAD-dependent?
Regarding the phenotype switching of TGFbeta-treated cholangiocytes, we removed this aspect statement as it is not pertinent to the focus of the our manuscript. SMAD proteins were not identified in our protein library, so we cannot make direct conclusion on their dysregulation. However, pathway analysis conducted on KEGG database highlights dysregulations in TGF-beta downstream pathways, which interaction with TGF-beta signalling is known to be mediated by SMAD proteins (i.e. cell cycle, apoptosis, ubiquitin mediated proteolysis). Thus a TGF-beta SMAD-dependent effect cannot be excluded, although more detailed studies would be needed to better elucidate this aspect.
- Conclusion: the formal conclusion section is absent. Moreover, the Conclusion paragraph in the end of the Discussion did not fully describe the findings of the present study.
According to your suggestion, we added the conclusion section where we summarized the results of our study.

Round 2
Reviewer 1 Report
- The quality of all the figures are very poor, it is very hard to read numbers presented in figure 1 C and 3.
According to your suggestions, the quality of all the figures was improved, increasing the resolution of the images to 800 dpi.
Comment: Appreciated
- Error bar and statistics are missing in figure 1C, similarly stats are missing in figure 2b.
Regarding figure 1C and 2b, the values between different conditions were analyzed by Kruskal–Wallis test and two-way ANOVA followed by Dunnett's multiple comparisons test, respectively, using GraphPad Prism software. P <0.05 was considered statistically significant. Statistical analysis showed no significant differences. We have elucidated these aspects in the captions, as you suggested.
Comment: If the data is non significant why are you claiming as positive, you need to remove the data if it is non significant.
- Line 199-200, authors claimed that the cytotoxic effect of TGF beta is not mediated by apoptosis, did they perform any experiment to prove the claim?
The cell cycle analysis does not show the presence of the sub G1 peak, which is indicative of cell apoptosis. Based on this we say that cytotoxic effect of TGF beta is not mediated by apoptosis.
Comment: I do not buy authors argument, claim must be supported by experiment.
- Authors have performed quantitative MS to identify differentially expressed proteins post TGF beta induction, what was the internal control and how many times the experiment was repeated. I would also perform validation of identified proteins by western.
Quantitative MS was performed analysing samples with the same protein content. Samples were prepared to achieve a final peptide concentration of 2 µg/µl before liquid chromatography–tandem MS (LC-MS/MS) analysis and 5 µl were injected, thus analysing a total amount of 10 µg of proteins for each sample. In addition, at post-processing data analysis, protein abundances were normalized based on the media total abundance per sample, making the different runs comparable with each other. Chromatographic performances and TOF accuracy were evaluated using an intra-run injection (5 µl) of beta-galactosidase 100 fmol/µl as internal control. LC-MS/MS analyses were repeated in triplicate for each sample.
We agree with the observation of the reviewer about validation by western, but we would underline our choice of LC-MS/MS approach. Sequential Window Acquisition of All Theoretical Mass Spectra (SWATH-MS) is a specific variant of data-independent acquisition (DIA) methods and has developed as a technology that combines deep proteome coverage capabilities with quantitative consistency and accuracy (please refer to following publications for details: L. C. Gillet et al., “Targeted data extraction of the MS/MS spectra generated by data-independent acquisition: A new concept for consistent and accurate proteome analysis,” Molecular and Cellular Proteomics, vol. 11, no. 6, 2012, doi: 10.1074/mcp.O111.016717; C. Ludwig, L. Gillet, G. Rosenberger, S. Amon, B. C. Collins, and R. Aebersold, “ Data‐independent acquisition‐based SWATH ‐ MS for quantitative proteomics: a tutorial ,” Molecular Systems Biology, vol. 14, no. 8, Aug. 2018, doi: 10.15252/msb.20178126).
Comment: I understand authors response, however, confirmation with additional data is needed.
- In the discussion line 284-85, authors are hypothesizing that the calcium signaling and concentration is perturbed after the treatment, I don't see any experimental proof of the claim, these kind of claim must be avoided without the proof.
We agree with reviewer about the need of an experimental proof. We would like only to say that our hypotheses are based on quantitative proteomic results. These proteomic data can paved the way for studies regarding the evaluation of energy metabolism and calcium homeostasis in TGF‐β-treated cholangiocytes to confirm the hypothesis.
Comment: Appreciated
Author Response
1. Error bar and statistics are missing in figure 1C, similarly stats are missing in figure 2b.
Regarding figure 1C and 2, the values between different conditions were analyzed by Kruskal–Wallis test and two-way ANOVA followed by Dunnett's multiple comparisons test, respectively, using GraphPad Prism software. P <0.05 was considered statistically significant. Statistical analysis showed no significant differences. We have elucidated these aspects in the captions, as you suggested.
Comment: If the data is non significant why are you claiming as positive, you need to remove the data if it is non significant.
We agree with the reviewer on the need for a statistical evaluation and indeed the differences we observed were not reported as statistically significant in the figure 1C and Figure 2b data and associated results sections. A brief description of these data reported in the text is based on the average value, since we cannot exclude that a small effect may still have biological relevance. In addition, it is worth pointing out that it was not possible to further increase the number of samples in order to make the statistics more robust as we are working with primary cells and not with immortalized cells.
2. Line 199-200, authors claimed that the cytotoxic effect of TGF beta is not mediated by apoptosis, did they perform any experiment to prove the claim?
The cell cycle analysis does not show the presence of the sub G1 peak, which is indicative of cell apoptosis. Based on this we say that cytotoxic effect of TGF beta is not mediated by apoptosis.
Comment: I do not buy authors argument, claim must be supported by experiment
Since to discuss cell apoptosis is not the focus of our paper, according to the reviewer suggestion, we modified manuscript from line 215 to 218.
3. Authors have performed quantitative MS to identify differentially expressed proteins post TGF beta induction, what was the internal control and how many times the experiment was repeated. I would also perform validation of identified proteins by western.
Quantitative MS was performed analysing samples with the same protein content. Samples were prepared to achieve a final peptide concentration of 2 µg/µl before liquid chromatography–tandem MS (LC-MS/MS) analysis and 5 µl were injected, thus analysing a total amount of 10 µg of proteins for each sample. In addition, at post-processing data analysis, protein abundances were normalized based on the media total abundance per sample, making the different runs comparable with each other. Chromatographic performances and TOF accuracy were evaluated using an intra-run injection (5 µl) of beta-galactosidase 100 fmol/µl as internal control. LC-MS/MS analyses were repeated in triplicate for each sample.
We agree with the observation of the reviewer about validation by western, but we would underline our choice of LC-MS/MS approach. Sequential Window Acquisition of All Theoretical Mass Spectra (SWATH-MS) is a specific variant of data-independent acquisition (DIA) methods and has developed as a technology that combines deep proteome coverage capabilities with quantitative consistency and accuracy (please refer to following publications for details: L. C. Gillet et al., “Targeted data extraction of the MS/MS spectra generated by data-independent acquisition: A new concept for consistent and accurate proteome analysis,” Molecular and Cellular Proteomics, vol. 11, no. 6, 2012, doi: 10.1074/mcp.O111.016717; C. Ludwig, L. Gillet, G. Rosenberger, S. Amon, B. C. Collins, and R. Aebersold, “ Data‐independent acquisition‐based SWATH ‐ MS for quantitative proteomics: a tutorial ,” Molecular Systems Biology, vol. 14, no. 8, Aug. 2018, doi: 10.15252/msb.20178126).
Comment: I understand authors response, however, confirmation with additional data is needed.
Data literature regarding behavior and signaling pathways of human cholangiocytes are scarce. With our study we wanted to lay the basis for deepening the information on this cell type. Proteomic analysis conducted aimed to evaluate at broad-spectrum the possible dysregulation of biological processes and the macro compartments (e.g. nucleus or plasma membrane) affected by TGF-beta, rather than focusing on individual proteins. Moreover, for proteins that could be interesting to validate such as U4/U6 small nuclear ribonucleoprotein Prp31 and nipsnap 1, antibodies are not easily available, their specificity should be verified and since they are poor expressed proteins WB results are uncertain. As already discussed, our LC-MS/MS approach combines deep proteome coverage capabilities with quantitative consistency and accuracy, making the validation with WB, that is considered a semi-quantitative analysis, superfluous in our case.

Reviewer 2 Report
After careful examination of the reviewers' comments, the response of the authors, and the changes made in the manuscript, I gather that the revised version of the manuscript has addressed some of the major concerns raised in the previous version of the paper. I am therefore happy to endorse the publication of this paper in the journal after language editing.
Author Response
We want to thank you for the suggestions and comments that have greatly improved our manuscript.

Round 3
Reviewer 1 Report
I still believe that the quality of paper could have been better if my reservation would have been addressed, however, this manuscript can provide an insight at the biological effect of TGF beta on primary cholangiocytes which is scarce indeed. I am satisfied with the arguments and explanation of the authors. I could see the improvement in the manuscript after revision and would like to congratulates authors for this work.